# Construction of an Electrochemical Receptor Sensor Based on Graphene/Thionine for the Sensitive Determination of β-Lactam Antibiotics Content in Milk

**DOI:** 10.3390/ijms21093306

**Published:** 2020-05-07

**Authors:** Lei Wang, Liyun Zhang, Yuke Wang, Yahong Ou, Xu Wang, Yuanhu Pan, Yulian Wang, Lingli Huang, Guyue Cheng, Shuyu Xie, Dongmei Chen, Yanfei Tao

**Affiliations:** 1National Reference Laboratory of Veterinary Drug Residues (HZAU) and MAO Key Laboratory for Detection of Veterinary Drug Residues, Wuhan 430070, China; leibest123@163.com; 2MOA Laboratory for Risk Assessment of Quality and Safety of Livestock and Poultry Products, Wuhan 430070, China; zhangliyun@webmail.hzau.edu.cn (L.Z.); wyk2095454314@163.com (Y.W.); oyh123999@163.com (Y.O.); wangxu@mail.hzau.edu.cn (X.W.); panyuanhu@mail.hzau.edu.cn (Y.P.); wangyulian@mail.hzau.edu.cn (Y.W.); huanglingli@mail.hzau.edu.cn (L.H.); chengguyue@mail.hzau.edu.cn (G.C.); snxsy1@126.com (S.X.); chendongmei@mail.hzau.edu.cn (D.C.)

**Keywords:** β-lactam antibiotics, electrochemical receptor sensor, graphene/thionine, milk

## Abstract

In antibiotics, β-lactam is one kind of major concern acknowledged as an unavoidable contaminant in milk. Thus, a facile and sensitive method is essential for rapid β-lactam antibiotics detection. In our work, a specific electrochemical receptor sensor based on the graphene/thionine (GO/TH) composite was established. The mechanism of the electrochemical receptor sensor was a direct competitive inhibition of the binding of horseradish peroxidase-labeled ampicillin (HRP-AMP) to the mutant BlaR-CTD protein by free β-lactam antibiotics. Then, horseradish peroxidase (HRP) catalyzed the hydrolysis of the substrate hydrogen peroxide (H_2_O_2_), which produced an electrochemical signal. Under optimal experimental conditions, this method could quantitatively detect cefquinome from 0.1 to 8 μg L^−1^ and with the limit of detection (LOD) of 0.16 μg L^−1^, much lower than the maximum residue limit (MRL) of 5 μg L^−1^ set by the European Union. In addition, the LOD of spiked milk samples with cefalexin, cefquinoxime, cefotafur, penicillin G and ampicillin were 14.88 μg L^−1^, 2.46 μg L^−1^, 17.16 μg L^−1^, 0.06 μg L^−1^, 0.21 μg L^−1^ and the limits of quantitation (LOQ) were 36.09 μg L^−1^, 5.40 μg L^−1^, 41.45 μg L^−1^, 0.13 μg L^−1^, 0.42 μg L^−1^, respectively. The sensor showed a favorable recovery of 84.89–102.44%. Moreover, the electrochemical receptor sensor was successfully applied to assay β-lactam antibiotics in milk, which showed good correlation with the results obtained from liquid chromatography-tandem mass spectrometry (LC-MS/MS).

## 1. Introduction

Notably, β-lactam antibiotics are the most widely used antimicrobial agents in the prevention and treatment of bacterial infectious diseases in animals, and are useful in the control of mastitis in dairy cows, urethral diseases, gastroenteritis and respiratory infections in other animals. In addition, they can be used as feed additives to prevent diseases in livestock and poultry, via the inhibition of bacterial cell walls. They play an important role in killing pathogenic microorganisms, and have the advantages of strong antimicrobial ability, bactericidal effects [1], low toxicity and broad spectrum [2]. However, due to the improper use, abuse and non-compliance of the withdrawal period, the existence of veterinary drug residues in animal derived foods such as milk and animal tissues poses a danger to human health, the ecological environment and food safety [3]. Further, the abuse of antibiotics may cause allergic reactions, microbial resistance and the general decline of immunity [4]. The Ministry of Agriculture of China establishes that the maximum residue limits of cefalexin, cefaquinoxime and ceftiofur in milk are 4–100 μg Kg^−1^ and 50–1000 μg Kg^−1^ in muscle, liver and kidney. EU Regulation 96/23/EC also establishes the maximum residue limits of β-lactam antibiotics in different animal tissue samples and the maximum residue limits (MRLs) of penicillin G and ampicillin in milk and muscle tissues are 4 and 50 μg Kg^−1^, respectively. These factors underscore the importance of the development of a rapid, simple, accurate and sensitive method for the detection of β-lactam antibiotics to ensure food safety [5,6].

At present, various available methods for β-lactam antibiotics detection in milk have been reported, mainly including sensitive liquid chromatography-tandem mass spectrometry (LC-MS/MS) [7], high-performance liquid chromatography (HPLC) [7,8] and enzyme-linked immunosorbent assay (ELISA) [9], but they demand expensive instruments, and there are frequent false positive results. In recent years, the development of biosensors has also progressed. Electrochemical analysis is considered as a green, highly sensitive and low-cost method for the detection of small biological molecules [10]. Additionally, while maintaining biocatalytic activity, nanomaterials have been used in the development of biosensor performance due to their ability to immobilize biomolecules, their large specific surface area, excellent biocompatibility and electrocatalytic properties [11]. Li and colleagues [12] developed a sensor based on a nanowire/nanoparticle hybrid array, which was used to simultaneously detect penicillin and tetracycline. Zhizhong Han modified the electrode with Au/ZnO/GN (PH-AZG), using penicillinase-hematoxylin oxide as a biological element, and successfully constructed a current-mode sensor of penicillin with excellent selectivity and stability. The sensor was able to detect residues at low concentrations in milk samples [13]. Therefore, a sensor based on carbon nanocomposites has achieved good results regarding the detection of antibiotic residues [14]. Scientific reports on the technology of biosensors mainly focused on the application of immunosensors and enzymesensors, while there are few research reports on the electrochemical receptor sensors [15]. The receptor sensor assay is a method that has been developed recently. It is based on the specific affinity that can occur between small biological molecules.

The penicillin binding protein (PBP) BlaR is a signal transduction membrane protein, which induces the synthesis of β-lactamase. The C-terminal domain of BlaR (BlaR-CTD) protein is located in the extracellular region, which acts as a drug binding site [16]. BlaR-CTD is firstly the sensor domain of a penicillin receptor that is acylated by penicillin. This protein can identify and bind to a variety of β-lactam antibiotics [17,18]. In the active site of the protein, STYK, serine is a key amino acid (AA) that can participate in the binding of β-lactams [19]. Thus, BlaR-CTD protein has been used to detect the β-lactam antibiotics residues in the receptor-based screening assay, such as a biological fluid method using colloidal gold labeled receptor protein [20] and a receptor-based enzyme linked immune-sorbent assay developed by our lab, in which BlaR-CTD from B. licheniformis ATCC14580 was immobilized on the plate [21]. Studies have shown that different types of penicillin binding protein (PBP) BlaR have different affinity of each β-lactam antibiotics [22]. The BlaR-CTD protein from Streptococcus pneumoniae can be used to develop a microplate assay, in which the assay was based on competitive inhibition of the binding of horseradish peroxidase-labeled ampicillin (HRP-AMP) to the PBP3 by free β-lactam antibiotics in milk [23]. In addition, Cacciatore G developed an assay to detect the residues of penicillins and cephalosporins in milk using a surface plasmon resonance (SPR) biosensor. The assay was based on the inhibition of the binding of digoxigenin-labelled ampicillin (DIG-AMPI) to a soluble penicillin-binding protein 2x derivative of Streptococcus pneumoniae [14].

Owing to the thermal instability and low affinity of BlaR-CTD to some β-lactam antibiotics, the receptor assay based on BlaR-CTD was limited in the detection of abundant variety of drugs and the result was often unstable [15]. So, in our lab, the study for the first time modified BlaR-CTD protein by rational design and site-directed mutagenesis, and the mutant protein I188K/S19C/G24C exhibited not only the improvement in stability, but also had a higher affinity than the wild-type [24]. In subsequent studies, mutant protein I188K/S19C/G24C was used to establish a fast and effective receptor method for the screening of β-lactam antibiotics residues.

Herein, we modified the glassy carbon electrode (GCE) through the deposition of graphene and thionine composites to realize signal amplification. On the basis of the modified electrode, we further developed a novel electrochemical receptor sensor with the mechanism of a direct competitive inhibition of the binding of horseradish peroxidase-labeled ampicillin (HRP-AMP) to the mutant BlaR-CTD protein by free β-lactam antibiotics. Finally, the electrochemical reaction signal was obtained via efficiently catalytic activity of horseradish peroxidase (HRP) towards enzyme substrate H_2_O_2_, in which thiophene was used as an effective electron transfer mediator [25,26]. In this paper, we researched the electrochemical behavior of electrodes modified using cyclic voltammetry (CV) and electrochemical impedance spectroscopy (EIS) [27]. The aim of this study is to provide a sensitive and rapid electrochemical analytical method for the quantitative detection of cefalexin, cefaquinoxime, ceftiofur, penicillin G and ampicillin in milk samples.

## 2. Results and Discussion

### 2.1. Purification of Protein

The induction of protein expression was carried out according to previous studies in our laboratory [24]. The optimal induction condition for the expression of protein was IPTG at a final concentration of 1 mM and culturing for 12 h at 18 °C. The protein was almost exclusively found in the supernatant after ultrasonic fragmentation, suggesting that conditions used here could be applied to induce the expression of subsequent proteins. The purified wild-type and mutant proteins were shown in Figure 1; each single point was purified with purity above 95%, indicating that the protein was purified successfully. The band for mutant protein I188K/S19C/G24C was in good agreement with the exoected size of 26 kDa. After the determination of protein concentration, 0.8 mg mL^−1^ of purified mutant protein was stored at −20 °C for further use.

### 2.2. Identification of Enzyme Markers

The ultraviolet spectrum of the HRP, AMP and HRP-AMP were shown in Appendix A. The results showed that the ultraviolet absorption spectrum of HRP-AMP (λmax, 403 nm) was consistent with that of HRP (λmax, 403 nm), which revealed that there was no change in enzyme activity. Moreover, the ultraviolet absorption spectrum of the coupling was also deviated from HRP to some extent. Therefore, the HRP-AMP was successfully prepared and the concentration of HRP-AMP was 0.36 mg mL^−1^.

### 2.3. Characterization of GO/TH Composite Materials

The graphene thionine composite was also characterized by ultraviolet-visible spectrophotometry, as shown in Appendix A. The curve a depicted the ultraviolet-visible absorption spectrum of thionine solution; it had a strong absorption peak at 601 nm. In the curve b, the graphene had no absorption peak at 601 nm. Curve c was the ultraviolet-visible absorption spectrum of the graphene thionine composite; the absorption peak was similar to that of curve a at 601 nm. These results indicated that the graphene/thionine (GO/TH) had been prepared successfully.

### 2.4. Characterization of the GO/TH/GCE

In this research, both CV and EIS methods were adopted to test the interface features of the stepwise modified receptor sensor. Figure 2 displayed the Nyquist curves of 5 mM [Fe(CN)6]^3−/4−^ containing 1 M KCl at stepwise established electrodes. Afterwards, the bare GCE electrode displayed a quite small semicircle domain, indicating a quite fast electron-transfer process (curve a), which was characteristic of a mass diffusion limiting step in the electron-transfer process. Because of the conductivity of GO/TH, an apparent decrease of semicircle diameter was achieved after the step of electrochemical deposition (curve b). The built film contributed to promote the electron transfer rate of the [Fe(CN)6]^3−^/[Fe(CN)6]^4−^ couple and obtained a larger interface area, herein improving the loading amount of BlaR-CTD. However, a stepwise growth in semicircle diameters corresponding with the increased resistances happened after the sequential fabricating of BlaR-CTD and BSA (curves c and d). The reason for that was the protein layer acting as the inert electron, while mass-transfer blocking layer significantly deterred the diffusion of ferricyanide towards the electron surface. The above EIS results indicated that electrochemical receptor sensor had been successfully constructed [28,29,30,31]. Figure 3 showed the CV results of different fabricated electrodes performd in 5 mM [Fe(CN)6]^3−/4−^, containing 1 M KCl at 50 mV/s scan rate. When the GO/TH deposited onto the GCE surface, the redox peak current went up dramatically (green curve), indicating that the GO/TH could accelerate electron transfer speed. In sequence, when receptor protein (BlaR-CTD) was dropped onto the activated electrode surface, apparently the peak current decreased, owing to the steric hindrance and an obstacle of insulated BlaR-CTD for the electron transfer, explaining that BlaR-CTD was successfully immobilized on the GO/TH/GCE surface (purple curve). After the receptor sensor was incubated with BSA, a further decrease in peak current was produced for the increasing of electron transfer resistance (red curve), implying that we successfully obtained the fabricated electrochemical biosensor for binding with BSA [28,32].

### 2.5. Optimization of the Detection Conditions

In order to best analyze β-lactam antibiotics, we optimized several key experimental parameters. CV was used for the detection of protein and HRP-AMP, due to its high sensitivity and good resolution. After the incubation of the protein in different concentrations, the CV signals of them were recorded in phosphate buffered solution (PBS) and H_2_O_2_ mixture at 50 mV/s scan rate under the optimal experiment conditions. The dilution ration of mutant protein I188K/S19C/G24C and HRP-AMP greatly affected the sensitivity and enzymatic reaction rate. In Figure 4, the CV response gradually increased with the reducing dilution ratio of mutant protein I188K/S19C/G24C from 1:3200 to 1:200 and gradually trended to level off at 1:800. Thus, 1:800 (1 μg mL^−1^) was used in the experiment as the optimal mutant protein ratio. Similarly, as displayed in Figure 5, the CV response gradually increased with the reducing dilution ratio of HRP-AMP from 1:2400 to 1:150 and gradually trended to level off at 1:600. Thus, 1:600 (0.6 μg mL^−1^) was chosen as the best optimal ratio.

Figure 6 showed the effect of the different pH, incubation time and incubation temperature on the current response of the CV sensor, which performed in phosphate buffered solution (PBS) and H_2_O_2_ mixture at 50 mV/s scan rate. The PH value of PBS plays a crucial role in electron transfer rate and influences the structure and function of the HRP protein. We prepared PBS with different PH values (5, 6, 7, 8) to explore the effect of PH on the enzyme activity. As shown in Figure 6A, the response current kept a stable value at 6–8. Considering the stability of polythionine, we fixed the best PH value as 6 to investigate the incubation time. Figure 6B showed the relationship between the incubation time and the corresponding response peak current. We recorded the peak current, which increased to a relatively stable value after a 45 min incubation period. Therefore, we selected 45 min as the optimal time. In addition, the incubation temperature is also used as a useful method to improve the detection sensitivity. As shown in Figure 6C, in the first 50 °C, the peak currents increased gradually. However, the peak currents gradually decreased with the further increase of incubation temperature. Similarly, considering the enzyme activity, sensitivity and service life of the electrode, we selected 37 °C as the best incubation temperature.

### 2.6. Linear Range and LOD

To investigate the performance of this electrochemical receptor sensor, different concentrations of cefquinome accurately prepared with PBS were tested, and the resulting CV responses were shown in Figure 7. Peak current values increased linearly as the logarithm of cefquinome concentrations increased from 0.1 to 8 μg L^−1^, producing a correlation coefficient of 0.9946 (Figure 7). The error bars indicated the standard deviation calculated from three independent measurements.

The limit of detection (LOD) of the sensor was calculated using a linear regression curve based on the method reported in the literature [29,33]. LOD is calculated as follows: LOD = 3 × Sa/b, where Sa is the standard deviation of the intercept and b is the slope of the calibration curve. Using the linear regression y = −66.745 *×* l g (cefquinome) (µg L^−1^) + 117.06, presented in Figure 7, the calculated LOD value for cefquinome was 0.16 μg·L^−1^ (linear range: l g 0.1 (μg L^−1^) – l g 8 (μg L^−1^), R^2^ = 0.9946).

As shown in Table 1, electrochemical receptor analysis using BlaR-CTD as bio-recognition element had a higher recovery rates when compared with ELISA. The method showed a stronger anti-matrix interference ability and provided a higher accuracy rate.

### 2.7. Reproducibility, Repeatability and Stability of the Sensor

To evaluate the reproducibility and repeatability of this developed electrochemical receptor sensor, we carried out five reduplicative tests of cefquinome at six different concentrations (0.1, 0.5, 1, 2, 4, 8 µg L^−1^), with the same batch of sensors. As shown in Appendix A, the coefficients of variation were less than 12.4%, which indicated that the sensor possessed a high level of reproducibility and repeatability. The stability was evaluated by recording CV current signals of cefquinome, after storing the electrode at 4°C for 14 d. After two weeks, the current response retained more than 91.31% of its initial response, which indicated that the system possessed a high degree of stability. Taken together, these results indicated that GO/TH/GCE electrochemical receptor sensors had excellent reproducibility and stability, which made the sensor a promising tool for the quantitative analysis of β-lactam antibiotics.

### 2.8. Real Sample Analysis

To detect the content of five major β-lactam antibiotics in milk samples, the CV current values for 20 blank milk samples were obtained using an electrochemical receptor analysis. The CV current values were brought into the regression equation to calculate the detection value of blank milk samples. The average (C) and standard deviation (SD) of 20 blank milk samples were sequentially calculated. According to the formula Z = C + 3/10 SD, the LOD and limits of quantitation (LOQ) values of cefalexin, cefaquinoxime, ceftiofur, penicillin G, and ampicillin in milk samples were obtained (Table 2).

To validate the applicability of the presented method for sensitively detecting those five kinds of β-lactam antibiotics (cefalexin, cefquinoxime, ceftiofur, penicillin G and ampicillin), the spiked milk at concentrations of 1 × LOQ, 2 × LOQ and 4 × LOQ were assessed with the suggested method. As shown in Appendix A, the recovery in milk samples was 84.89–102.44%. The coefficients of variation were less than 11.34%. These results indicated that this electrochemical receptor sensor could be applicable for the detection of β-lactam antibiotics in food matrices.

## 3. Materials and Methods

### 3.1. Materials

Cefalexin and Ampicillin were purchased from the China Institute of Food and Drug Verification (Beijing, China). Cefquinome, Ceftiofur and Penicillin G were provided by Dr. Ehrenstorfer (Deisenhofen, Germany). BlaR-CTD receptor protein had been developed in our laboratory. The NI-NTA affinity nickel column was purchased from GE Company, Waltham, MA, USA. BCA-100 Protein Quantitative Detection Kit (Enhanced), Tris-HCl (1.5 M, pH 8.8) and Tris-HCl (1.0 M, pH6.8) were obtained from the Biyuntian Instituie of Biotechnology. EDTA, Kanamycin, NHS, EDC, HRP and BSA were purchased from Beijing Solebo Technology Co., Ltd. (Beijing, China). IPTG was purchased from Anhui Biosharp Company (Anhui, China). All other chemicals, such as sodium dihydrogen phosphate and potassium dihydrogen phosphate, were of analytical grade and were purchased from Sinopharm Chemical Reagent Co., Ltd. (Shanghai, China).

### 3.2. Apparatus

All the electrochemical experiments including CV and EIS were performed with the Autolab PGSTAT128N (Vanton, Switzerland). All measurements were carried out using a conventional three-electrode system that was composed of Ag/AgCl, which was saturated with KCl as reference electrode, a platinum wire as auxiliary electrode and a GCE with a diameter of 3 mm that was modified with the graphene/thionine working electrode at room temperature. The scanning electron microscope images and atomic force microscope images were photographed using a SU8010 scanning electron microscope (Hitachi, Japan) and Multimode 8 (Cambridge, MA, USA).

### 3.3. Preparation and Purification of Protein Samples

*E. coli* BL21 (DE3) harboring mutant plasmid were cultured overnight in 1 L of LB broth containing 50 μg mL^−1^ kanamycin at 37 °C in shaker. The culture was diluted 100-fold into a fresh LB broth containing 50 μg mL^−1^ kanamycin and incubated with vigorous shaking at 37 °C, until the OD600 reached 0.6. Then, 1 mM IPTG was added and the culture was incubated at 18 °C for 12 h. Bacteria were collected by centrifugation at 12,000 rpm for 10 min, and were washed 3 times with ice cold PBS. Then, 50 mL binding buffer (7.6 g Na_3_PO_4_, 29.22 g NaCl, 0.68 g imidazole per liter, pH 7.4) was added into the precipitate, and ultrasonic fragmentation was carried out after resuspension of the precipitate. The suspension was centrifuged at 12,000 rpm for 40 min at 4 °C, and the soluble fraction was loaded onto a 2 mL Ni^2+^-charged chelating sepharose resin column. After washing by different concentrations of imidazole eluent, the protein was eluted at an imidazole concentration of 60, 100 and 200 mM. The purified mutant proteins were dialyzed against PBS, and were confirmed by SDS-PAGE with Coomassie Brilliant Blue staining and Western blot [24].

### 3.4. Preparation of Enzyme Markers

Ampicillin was coupled to HRP, as described by Zhang J et al., with some modifications [23]. Briefly, 2.4 mg of EDC (1-Ethyl-3-(3-dimethylaminopropyl) carbodiimide) dissolved in 1 mL of PBS was added dropwise to 6.18 mg of ampicillin, followed by 1.8 mg of NHS (N-Hydroxysuccinimide) in 1 mL of PBS. The reaction mixture was stirred for 2 h at room temperature. Then, 4.3 mg of HRP dissolved in 2 mL of PBS was added dropwise to the above mixed solution, and stirred for 10 h at 4 °C in dark. Subsequently, dialysis against 2 L of PBS was performed over 3 days. Finally, we tested the UV-vis spectrometry of HRP, AMP and HRP-AMP, to identify the coupling result.

### 3.5. Preparation and Characterization of GO/TH

To prepare the graphene thionine composites, graphene was dispersed in PBS solution and 2 mL thionine solution (5 mg mL^−1^) was then added. The mixture was agitated at room temperature for 24 h, after which the composite was centrifuged. After being washed with ultra-pure water, the graphene thionine composite was stored in PBS solution at 4 °C. The GO/TH was characterized by UV-vis spectrometry.

### 3.6. Preparation of Electrochemical Receptor Sensor

The electrochemical receptor sensor was prepared by the following steps. (i) The GCE was polished by α-Al_2_O_3_ powder with different sizes (by using them with a size of 0.3 µm and 0.05 µm in sequence). GO/TH/GCE was prepared by drop-casting of GO/TH suspension onto the GCE. (ii) Then, we immediately dropped 10 μL of 1 µg·mL^−1^ BlaR-CTD protein onto the electrode surface and incubated it for 45 min at 37 °C. (iii) Following this, we rinsed the built electrode with PBS and next treated it with 10 μL of 1% BSA solution for 30 min at 37°C to block the inactive sites. (iv) Finally, we thoroughly rinsed the electrode with PBS again. The electrochemical receptor sensor was successfully prepared and stored at 4 °C [32,34]. The fabrication process for this electrochemical receptor sensor was displayed in Scheme 1.

### 3.7. Electrochemical Measurements

In order to perform the competition assay, we firstly blended 10 µL of the diluted HRP-AMP (0.6 µg·mL^−1^) with 10 µL of antibiotic standards solution, with different concentrations from 0.1 to 8 µg·L^−1^ in PBS. We added the miscible liquids onto the modified BlaR-CTD protein electrochemical receptor sensor surface and then incubated it at 37 °C for 45 min. In the incubation process, a competitive reaction happened between the immobilized HRP-AMP and free AMP toward a rationed BlaR-CTD protein on the electrode. Then, we washed the resulting electrode with PBS and finally immersed it into the freshly prepared phosphate buffered solution containing hydrogen peroxide. The potential cycling range of cyclic voltammetry (CV) was from −0.2–0.8 V; the scanning rate was 50 mV/s.

### 3.8. Sample Preparation

The whole milk was purchased from the local supermarket. We took 10 mL of every sample into 50 mL tubes, and added 1.718 g ammonium sulfate, and then centrifuged at 8000 rpm for 10 min at 4 °C after a 5 min vortex. After that, the supernatant solution was filtered by 0.22 μm filter membrane. In sequence, adjusted the PH value of the solution to 6 for electrochemical receptor sensor analysis.

## 4. Conclusions

In conclusion, we developed a novel electrochemical receptor sensor based on GO/TH/GCE nanocomposites, with the high affinity mutation BlaR-CTD protein as a specific bio-recognition element used for the rapid detection of β-lactam antibiotics in milk. The signal amplification strategy on the GO/TH prominently improved the sensitivity of β-lactam antibiotics determination. Simultaneously, the HRP possessing excellent catalytic activity towards H_2_O_2_ enhanced the sensitivity of this assembled receptor sensor. The developed receptor sensor was successfully applied to detect five major β-lactam antibiotics, and overcame the challenge caused by the low recognition of a single drug to a certain extent. For cefquinome, it obtained a linear relationship at a concentration range from l g 0.1 (µg L^−1^)–l g8 (µg L^−1^); the detection limit was 0.16 µg L^−1^. Moreover, the sensor showed excellent reproducibility, high sensitivity and accuracy, and might also be applicable to other kinds of β-lactam antibiotics. The system is effective and reliable, has a wide range of potential applications and may be ensured to guarantee food safety in the future.

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
