# Peer review of "Construction of an Electrochemical Receptor Sensor Based on Graphene/Thionine for the Sensitive Determination of β-Lactam Antibiotics Content in Milk"

_ijms, 2020, doi:10.3390/ijms21093306_

Round 1

Reviewer 1 Report

  The reviewed work entitled “Construction of an electrochemical receptor sensor based on graphene/thionine for the sensitive determination of β-lactam antibiotic content in milk” (ijms-765990) does not need any special comment. The risen topic is original and important considering development of novel electrochemical sensors as well as an innovative, highly sensitive methods for β-lactam types of antibiotics determination in milk and edible meat samples. From this vital viewpoint the consideration of the manuscript publication is worth to be undertaken in my opinion.

  From the manuscript itself, I can assumed that the Authors were well prepared to perform described experiments and their laboratory provided all necessary equipment to performed reliable research. This conclusion is an outcome of the number and quality of the obtained measuring data, which are sufficient for preparation of a good manuscript. The literature part is interesting and prepared properly same as the cited articles that are chosen correctly with proper justification. The discussed topic is undoubtedly up-to-date and important. I think, that the work fulfils the basic novelty criteria of the International Journal of Molecular Sciences.

  However, the paper should be revised to provide this clarification before I can recommend it publication. The significant misinterpretation of obtained results leading to false conclusion is my general objection to the manuscript. As an example lets analyze the sentence taken from the Conclusion part: “In the concentration range of 0.1–8 ug L-1, the linear equation was I(uA) = -66.745 × l g (cefquinome) (ug L-1) + 117.06, the correlation coefficient was 0.9946, and the sensitivity was 0.16 ug L-1.” There are three substantial mistakes done in just one sentence.

a) The Authors provides “concentration range of 0.1–8 ug L-1” whereas in Fig. 7 the logarithm of cefquinome concentrations is presented, which undermine the credibility of the whole work!!!

b) „I(uA) = -66.745 × l g (cefquinome) (ug L-1) + 117.06” the drug concentration unit is ubiquitous

c) “sensitivity was 0.16 ug L-1.” provided value is LOD!!

  Furthermore, there are countless substantive errors like: (Line 289: "Cyclic voltammetry was conducted at a voltage of 0.20.8 V” in fact it was a range from -0.2 to 0.8 V and grammatical errors as well (native speaker revision of text is recommended) throughout the manuscript.

  Strong sides of the work: well planned and executed experiments, presented results are obtained by high-quality equipment. Weak sides of the work: week interpretation of the measurements results.

  Summarizing, in my opinion the work in its current version cannot be published in the International Journal of Molecular Sciences. But I am ready to reconsider it as soon as the Authors improve the manuscript  quality and present logically, that the method can guarantee such perfect results.

Reviewer 2 Report

The manuscript describes the construction of an electrochemical receptor sensor based on graphene/thionine for the sensitive determination of β-lactam antibiotic in milk.

The title is not in line with the content. The sensor detect several antibiotics, not just one.

In the abstract (line 31) is stated “meat food.” It is a mistake?

The detection scheme presented in the scheme is not in accord with the title and the abstract of the manuscript. Furthermore, it not comprehensive.

The voltammetric studies presented in the Figure 3 are not described and discussed. Pleas improve.

The measurement of the current in the optimization studies is not described. What technique was used for?

The results obtained in the case of pH optimization are not logical. Change of two pH units do not reflect any modification for the sensor response? It is not possible. Please check and improve.

The calibration results are not clear. What technique was used for the measurements? What is the concentration range?

In the real samples CV was used as detection technique. The calculations are unclear. All the samples were spiked with the analyts. The question is: the sensor is able to detect the antibiotics at levels that these could be found in milks? It is not prove on it.

The selectivity of the sensor is not studied. It is very important the selectivity towards different antibiotics.

The preparation of the samples is too complicated. What it was necessary? Please explain.

Round 2

Reviewer 1 Report

  After a thorough revision of the work I assume, that it complies with journal requirements and can be published in the International Journal of Molecular Sciences.

  However, in every part of the manuscript text I still find lots of language mistakes that need to be corrected before publication.

  Nevertheless, the substantive part of the work does not rise any major objectives right now.

Reviewer 2 Report

The revision was well carried out and the paper could be published without further modification.

Author Response

Thank you for your comments concerning my manuscript. We have studied comments carefully and have done our best to correct the language errors. Revised portion are marked in red in the paper. The main corrections in the paper and the responds to the comments are as flowing:

List of Actions

Line 15: We changed “were” to “is”.

Line 35-48: We corrected some language mistakes and rephrased some sentences.

Line 52-69: We re-expressed some sentences.

Line 76, 78, 90: We changed “β-lactams” to “β-lactam antibiotics”.

Line 94-95: We re-expressed some sentences.

Line 111, 115,125,146,147,156: We corrected some language mistakes.

Line 171-172: We revised a reaction condition.

Line 190-196: We corrected some language mistakes and rephrased some sentences.

Line 207, 208,221,227,228,242: We corrected some language mistakes.

Line 290, 291,302,311,316,328: We corrected some language mistakes.

Once again, thank you very much for your comments and suggestions.